# Around Ruins: Some Notes on Feminist and Decolonial *Conversations* in Aesthetics

**Márcia Oliveira** 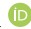

The Center for Humanities of the University of Minho (CEHUM), 4710-057 Braga, Portugal; marciacoliveira@gmail.com

**Abstract:** Although using different strategies, Portuguese artists Mónica de Miranda and Filipa César make us think about and reflect on the ruins of the Portuguese 'empire' but also on the ruins—and the remains—of European colonialism and its patriarchal backbone. Their work opens the possibility of discussing aesthetics from feminist and decolonial perspectives, departing from the category of 'ruins' and considering the many ways through which these ruins and their multiple inflections contribute to the creation of potential affective geographies and memories.

**Keywords:** ruins; Mónica de Miranda; Filipa César

## 1. Introduction

Although using different strategies, Portuguese artists Filipa César and Mónica de Miranda make us think about and reflect on the ruins of the Portuguese 'empire' but also on the ruins—and the remains—of European colonialism and its patriarchal backbone. Their work opens the possibility of discussing feminist and decolonial aesthetics departing from the category of 'ruins', considering the many ways these ruins and their multiple inflections contribute to the creation of potential affects across geographies and memories. Mónica de Miranda, and Filipa César resort to immaterial media (film, performance, and even photography to a certain extent) to problematize the post-independence moment in the territories occupied by Portugal during its colonial rule in Africa—media which, by their very nature, manage to capture the diversity of the ruins of that 'empire', its geographical dispersion, and its temporal scope: material ruins, archival ruins, epistemological ruins, and ruins of representation. Building upon these ruins, these practices emphasize how aesthetic value is not a predetermined essence of an artwork, its materials, and its processes, or how it is not dependent on the formulation of a subject. It is more a layering of historical, political, and cultural circumstances (both collective and individual) that are at play in forming a line of affects and sensitivity. These circumstances are thus being reconfigured by artists focusing on black bodies, post-colonial momentum, and critique and decolonial *conversations*.[1] The notes presented here try to establish potential connections between thinking and praxis from the perspective of the artists and their works, which cannot be reduced either to an artistic or an activist *praxis*, or even to a critical action of thinking beyond the limits and impositions of coloniality. Ruins, in their material, conceptual, and epistemological guises, emerge here as 'undisciplined' entities that entice potential *re-significations* (Mignolo and Walsh 2018)—something like "raw materials", that "refer to the past and present configurations of the global ( . . . ) context" (Ferreira da Silva 2018, p. 1).

### 1.1. Portuguese Dictatorship and Colonialism—A Brief Context

Even though Portuguese decolonization happened right after the military coup that ended the longest dictatorship in Europe in April 1974, the fact is that, as Ana Balona de Oliveira notes, "if political decolonization took place in 1974–1975, epistemic, psychic and institutional ones did not" (Oliveira 2018, p. 233). The myth of Portugal being a

"good colonizer" with a soft hand persisted and instilled a lingering collective fallacy that Portuguese colonialism was very different and more positive than other European counterparts, such as France, or the Netherlands, for example.[2] This myth has been one of the most discussed issues of Portuguese Colonialism in the past few decades, not just in academia but also in society, especially via the printed press.[3] This resulted, in *grosso modo*, in an all-encompassing feeling that there was no racism in Portugal and no issues to be resolved. Only very recently, in the wake of the emergence of post-colonial studies, Portuguese memories related to the then-called "Empire" started being thoroughly addressed. As Margarida Calafate notes, "After 25 April 1974 we would think, almost intuitively, that all would be widely discussed, but the truth is that themes such as the dictatorship, and much more so the empire, the Colonial War that ended it and the decolonisation that followed, with the 'return' of thousands of people to Portugal, always constituted a kind of silenced memory, an uncomfortable memory, difficult to assume and elaborate by the new regime" (Ribeiro 2020, p. 5).

Also, such a myth was part of the ideological construction promoted by the dictatorial regime, Estado Novo, created by António de Oliveira Salazar and temporally delimited by two *coups d'état*—28 May 1926 and 25 April 1974.[4] Since its inception, the regime was building a solid ideological ground on different fronts, sustained by three pillars: "Deus, Pátria e Família" ("God, Country and Family": "The trilogy of National Education"). The triad synthesized a ubiquitous State focused on the centrality of the dictator; a strong relationship with Catholicism (and the inherent power distribution between the religious institution and the secular state), the model of a nuclear family based on patriarchal values,[5] and the increased annihilation of the rights of women throughout these decades. It was ideologically, politically, and economically structured as a "Colonial Empire", isolated from Europe,[6] which was one of its most distinctive characteristics, besides institutionalized censorship and overall repression and, of course, the Colonial War.

### 1.2. Fighting for Independence vs. Fighting for Colonialism

Words matter and, as Grada Kilomba acutely notes in the introduction to the Portuguese translation of her book *Plantation Memories*, in Portugal, the decolonization of language is yet to take place:

> "I write this introduction precisely because of language: on the one hand because I find it mandatory to clarify the meaning of a series of terms that, when written in Portuguese, are revelatory of a deep lack of reflection and theorization of history and of colonial and patriarchal heritage present in the Portuguese language; on the other hand, because, I must say, this is a wonderfully elaborate translation, for it translates an entire book with an absence of terms that were already dismantled, or even reinvented, in other languages, such as English, or German. However, in Portuguese, they remain deeply anchored in colonial and patriarchal discourse, thus becoming extremely problematic" (Kilomba 2019, p. 18) (my translation).

Language, or the terms commonly used to refer to or to describe certain events, certain things, images, feelings, etcetera, is revelatory of the position in which we, as individuals, face the world and also of the ways that a collective chooses to look to the past or envisage the future. This is the case of the vast nomenclature that the War between the Portuguese and the African colonized peoples has assumed and still assumes. For the colonizer: Guerra do Ultramar, Guerra de África, Campanhas de África, Guerra Colonial. For the indigenous peoples: Guerra de Libertação Nacional, Guerra de Independência. Starting in 1961, the war between the Portuguese army and the African liberation movements of Angola (MPLA, UNPA, UPA, FNLA, UNITA), Mozambique (FRELIMO), and Guinea-Bissau and Cape Verde (PAIGC)[7] only came to an end between 1974 and 1975, after the Revolution that overthrew the Dictatorship on 25 April, following the military coup. After World War II, the Portuguese government did not come to terms with the beginning of the political decolonization initiated by the Bandung Conference (held by 29 African and Asian countries

in Indonesia from 18 April to 24 April). The government resisted dialog with the liberation movements to negotiate independence and maintained an intransigent position, and the reaction to the assault led by Angolan nationalists to the Luanda prison on 4 February 1961 was violent repression. Troops were sent to repress the protests 'quickly and strongly'.[8] The building of the 'good colonizer' myth now had a new facet: the Portuguese Colonial War was pictured as being radically different and far less brutal than any other similar conflicts, which was another notion of the New State propaganda that lingered far beyond the end of the regime itself.[9] It was only as late as 2022 that the Portuguese government apologized for one of the massacres perpetrated during the conflict. During an official state visit to Maputo, and facing the Mozambican president Filip Nyusi, Portuguese Prime Minister António Costa said in his speech, on 2 November: "In this year of 2022, almost 50 years after that terrible day on 16 December 1972, I cannot fail to evoke and bow down to the memory of the victims of the Wairimu massacre, an inexcusable act that dishonours our History".[10] From the many ruins left by colonialism, the war, and the independence process, the ruin of memory was one of the most pervasive and complex. As Rebecca Solnit states, "forgetting is the ruin of memory, its collapse, decay, shattering and eventual fading into nothingness" (Solnit 2007, p. 151). Artists today stand atop these many ruins of a colonial Empire constructed for over 500 years, spanning the whole globe, occupying African territories like Angola, Mozambique, Guinea-Bissau, and Cape Verde, but also Brazil (until the independence in 1822), Goa (liberated by the Indian troops in 1961), and Macao (for which official rule was transferred to China in 1999).

## 2. On Feminism, Femininity, Aesthetics, and Beyond

In the book *Novo Mundo: Arte Contemporânea no tempo da pós-memória*,[11] researcher and curator António Pinto Ribeiro reflects on several artists whose works he places in the context of post-memory.[12] Ribeiro identifies post-memory as a "condition" in which the artists work, for "post-memory is not an aesthetic condition, nor an artistic genre, but an objective and subjective condition affecting artists, the cultural time, the producers, critique, museography and the anonymous receiver of the works, who in turn needs to modify his mechanisms of reception" (Pinto Ribeiro 2021, p. 9). These artists are thus placed because their position is truly transnational and cosmopolitan, indelibly marking the European art scene, as Ribeiro states, given the entanglement of their personal and family histories with transatlantic geographies in the context of post-independence African and Asian countries. Theirs is also, I would add, a transhistorical position, intertwining the "past" history of colonialism with its present. That is—and recurring to Georges Didi-Huberman—its survival in so many different guises (Didi-Huberman 2002). However, not being an aesthetic category, this position does have a deep and lingering impact on aesthetics, reclaiming new and sometimes uncomfortable (for the white European-centered eye) *ways of seeing*, feeling, and engaging with the artworks in question. Inventing *new tools to dismantle the master's house* (Audre Lorde), or to dismantle the intricate relationship between colonialism and modernity itself, which Walter Mignolo and Catherine Walsh identify as the "colonial matrix of power". As they say, "coloniality is constitutive, not derivative of modernity. That is to say, there is no modernity without coloniality thus the compound expression: *modernity/coloniality*" (Mignolo and Walsh 2018, p. 4). This is an entwinement akin to the one identified by several discussions of feminism and aesthetics, addressed by several feminist authors such as Peg Brand, Carolyn Korsmeyer, Rita Felski, or Christine Battersby, but also by bell hooks, Trin T-Minh Ha and Adrian Piper (in this case as both artist and philosopher), who have, from early on, introduced the pressing issue of ethnicity in this discussion.

Indeed, aesthetics has been one of the fields of philosophy more resistant to feminist critical approaches since the 1970s, but feminism itself has also resisted entering the realm of aesthetics. A tense relationship is based on two main issues: first, the strong roots of aesthetics in Western philosophical tradition (and its dogmas) are not easily combined with the cultural critique made by feminism since the 1960s; second, aesthetics' conception

that any work of art has an intrinsic/autonomous value, and also a universal one, is hard to put together with feminist analyses, anchored as they are in social and historical methodologies. As Peggy Zeglin Brand and Carolyn Korsmeyer point out, "These two broad theses have been challenged repeatedly in the latter part of this century, both by feminists and postmodernists, and earlier by followers of Wittgenstein. Postmodernism's challenge is especially acute on the issue of the universality of aesthetic appreciation. It questions the notion of common subjectivity and hence undermines what is strongest about theories that delimit a distinct area of aesthetic consciousness: their demonstration of a common human faculty that binds all together and permits transcendence of cultural barriers" (Brand and Korsmeyer 1995, p. 7). Adding this universalism to notions like 'artistic genius' (so well scrutinized by feminist critique), one can understand that feminism's reactions to aesthetics would be mostly according to two lines of work: aesthetics' prompt rejection (following the latest relationship between art and politics that was disseminated in the past two decades) or the defense of a 'feminist aesthetic' whose paradigms could oppose the masculine and patriarchal point of view of traditional aesthetics, thus providing an alternative to it. There are indeed problems in this relationship, starting with the definition of an aesthetic as either feminist or feminine. As Rita Felski puts it: "There is no distinctive style, medium, or set of techniques common to the work of all feminists, let alone female, artists. Rather, one can point to a multiplicity of genres and forms that are employed by women across the fields of contemporary art practice" (Felski 1995, p. 442). This brings to the fore the question of essentialism, a strong long-term debate in feminist aesthetics, but also, as Belinda Edmonson pointed out, in the field of Black Aesthetics: "The polemics and political thrust of this line of feminist criticism strongly resembles arguments made by the Negritudinists of the 1930s for a pan-African aesthetic and, more particularly, arguments of the (male) black literary theorists of the 1960s and 1970s, who similarly sought to define that ephemeral thing-the "black novel"-through the concept of a diasporically all-inclusive "black aesthetics"" (Edmonson 1992).

Further, and as Carolyn Korsmeyer also stresses, feminism is responsible for a significant loss of trust in systematic theorization, mainly through suspicions concerning the validity of the foundations of taste and critical appreciation, a question that, also, has a foundational character in what constitutes the philosophy of art: "Its challenge to claims about universal human nature, its support for pluralist interpretations of art, and its own brand of scepticism regarding "the canon" of great works"" (Korsmeyer 1993, p. 199). If feminism is important to discredit such theoretical foundations, it has been so in the sense of propelling the grand changes that occurred in aesthetics in the past few decades, presenting new challenges to the concepts of art and cultural value. Neither a 'feminist aesthetic' nor an 'aesthetic of the feminine' (rejecting a fixed and impermeable, thus essential, identity association), it is a political aesthetics of multiplicity that can redefine the notion of beauty and the parameters in which it is framed, beyond the Kantian dichotomous aesthetic model of the beautiful and the sublime and his universality of aesthetics judgment (Kant 1998).

The search for a multiple, flexible model that rejects being 'a paradigm' mines a model of thinking based on essentialism, be it gendered or ethnic, but still cannot completely deviate from any discussion based on the subject. But maybe the question to be addressed in knowledge production is one of *positionality* that does not define 'a' subject with a set of stable and unfixed characteristics, but rather a set of experiences, memories, and points of view that influence, or determine, the subject's creativity and sensibility—not just 'Who is Speaking' (Minh-Ha 1992), or creating, but from which place, which individual and collective history. I would say something along the lines of Denise Ferreira da Silva, who has been arguing for a *black poetics* (Ferreira da Silva 2014), for "what distinguishes black women's creations is precisely how they refigure the creative itself. Instead of actualizations or effects of a separate and self-determined entity who draws from a given (presented as particular or universal) interior (essential) or exterior (causal) condition, they can (and perhaps should) be approached as everything else that is of the world; that is, as

re/de/compositions of the elementary constituents of all that happens and exists" (Ferreira da Silva 2018, p. 5).

In this line, what ruins, as "contemporary ways of appropriating the past" (Assmann and Conrad 2010), provides the field of aesthetics with a process of reinvention—a construction, an envisionment of potentialities built around the concatenation of imagination (i.e., future) and memory (i.e., past), for "ruins are part of the long history of the fragment, but the ruin is a fragment with a future" (Dillon 2011, p. 11). In the introduction to the book *Memory in a Global Age*, Aleida Assmann and Sebastian Conrad argue for a change of focus in the discussion of globalization from imagination, as sustained by Appadurai in his seminal *Grassroots Globalization and the Research Imagination* to memory (Assmann and Conrad 2010). Still, the editors recognize that "memory, of course, also requires imagination, but it rethinks the future in alliance with recasting the past" (Assmann and Conrad 2010, p. 1). In this work, Appadurai defines this idea of 'Grassroots globalization' as the "relation between pedagogy, activism, and research in the era of globalization" (Appadurai 2000, p. 3), taking it as key to imagination and the positive force of globalization that counters its problematic relationship with the capital globally. Thus, he states, "the imagination is no longer a matter of intellectual genius, escapism from ordinary life, or just a dimension of aesthetics. It is a faculty that informs the daily lives of ordinary people in myriad ways. It allows people to consider migration, resist state violence, seek social redress, and design new forms of the civic association, and a collaboration, often across national boundaries" (Appadurai 2000, p. 6).

### 3. On *Beauty*, Epistemological Ruins, and the Ruins of Representation

Mónica de Miranda, a Portuguese–Angolan artist living between Lisbon and Luanda, defines her interdisciplinary work, encompassing video, photography, installation, drawing, and sound as "urban archaeology and personal geographies": "t is about deconstructing those places and recreating new maps of interaction where that geography is put together by the personal relations we have with the world and, culturally, with other spaces. If I had to draw my own map it would have traces of the place where I was born, or the place where I grew up, the places where I lived or even of a far way place, the place of ancestors" (Macedo et al. 2022, p. 210). If we would thus look at de Miranda's work through this observation, we would say that her *positionality* is an unfixed one, moving geographically, but also temporally. In it, we grasp the movement of thousands of people that migrated from Africa to Portugal, escaping civil wars[13] and looking for better living conditions, but only finding hardship, and, more often than not, undignified treatment, epistemic racism that lingered throughout generations,[14] and a total absence of voice and places from where to speak.[15] In her doctorate *Home Sweet Sour Home*, de Miranda deals with her family memories by revisiting, or re-imagining though drawing the places where she herself and her family lived, namely her grandmother and her mother—places she knew or that she didn't know, but that were somehow in her memory—Portugal, Angola, Brazil, England, India . . . The film *Once Upon a Time* (2013)[16] resulted from this practice as a research process around houses, bodies (feminine bodies), dwellings, movement, and memory in which the personal meets the collective. Existing spaces, ruined spaces, encompass a geopolitical meeting that also includes the body, those that actually move in space, while the houses, the places, move in time via their own ruination process (Stoler 2013). "In that encounter, of course, the personal dimension found yet another space, which goes beyond the personal sphere. In Angola, it was the ruins of the city, those urban archaeologies, those memories of the spaces that were once a place and now are another one, they are colonial ruins, the wounds of the city and urban planning—I move from the space of the house to the space of the city itself" (Macedo et al. 2022, p. 211). The film shows different fragments of those spaces, sometimes being inhabited—airports, cities, buildings (inside and outside), different landscapes, and, very significantly, the sea and the movement of water recurrently surfacing, indifferent to the ruined ships stranded on the shore.[17] Everything is, in a way, simultaneously real and imagined, past and future,

mother and child, wound and healing, belonging and *unbelonging*, for, as the artist herself says, "between cities, houses, airports and roads, private rooms, family houses, hotels, places of private life, in places of my own memories but simultaneously set also in those 'nonplaces' belonging to no one".[18] This work, as in fact many of de Miranda's works, can be read not just as a narrative of time and place displacement but also as a deep visual reflection on the deep-seated entanglements between colonialism and capitalism, and how such entanglement finds itself ingrained in every personal and quotidian experience. M. Neelika Jayawardane makes a compelling argument about the relationship between the access to the act of photographing (requiring from older generations the economic means to register the family's memories) and of building memories around those images registered throughout generations to that feeling of belonging and to the possibility of creating a solid self-image. The author says: "Our collections of familial and familiar circles are conduits to self, even when they do not depict us directly. They help us re-establish familiarity, belonging, self, and connection to those who are no longer in our realm of contact, creating a bond between our capacity for visualising belonging and stability. These acts of looking are indicative of our psychological need for returning to pasts that are no longer physically or emotionally available" (Jayawardane 2018). But, Jayawardane stresses, these "images are not stable", even though their *lacunae* represent a "separation from 'normative' narratives of modernity [that] leaves us with an abundance of longing; we recreate returns, and build image banks that signify the contradictions of displacement—to establish belonging and right of return, and the impossibility of return" (Jayawardane 2018).

Landscapes (de Miranda focuses as much on natural landscapes as well as on city-scapes) and portraiture are the ultimate visual representations of the European project of modernity,[19] specifically in painting. However, the artist says, "the landscapes presented in my work [ . . . ] refer to a colonial legacy, but seen from a postcolonial world [ . . . ]. Colonial control over territory is well known and discursively and established through technologies of surveying, mapping and representing landscape. The landscapes at this juncture are living and geography of subjective spaces" (de Miranda 2017, p. 72). It is, thus, quite significant that de Miranda keeps going back to those genres in her films and photographs, placing the black body (also significantly dressed in white for most of the time) in the center of the representation space. This brings to the fore a process of ruination of the aesthetic autonomy of such genres (metonymically, of art and representation as well); of the still very much in debt to Western 'classic' conceptions of ideal beauty. In the 2018 short film *Beauty*,[20] we are shown a place of artistic learning (the Académie des Beaux Arts and the Tour de l'Échanger in Kinshasa, where part of the film was shot). In the first frame of the composition, we see a white stone sculpture of a black boy standing in a garden, his gaze oriented towards the sky, while in the second frame, another man's gaze, this time looking down towards the floor, is cast in the same material. This significant difference is paired with another interesting detail: the decaying of the second sculpture, with one arm broken, as if correlating with the weariness suggested by the way he keeps standing while everything else bears many signs of material and psychic destruction. These sculptures, carved in the same stone, should be compared to the stereotype of a woman, naked, in the fashion of the best Western art's objectification of the feminine body as the object of a masculine gaze (Mulvey 1975). Amidst these ruined sculptures (a material and symbolic ruin, for they are broken, not whole, and are thus desecrated as a valued object) stands a living woman's body in a draped white cloth (a model, for sure) that looks at them defiantly. The abandonment of the teaching space, of the anatomical chalkboard notes and drawings, refers back to the decaying of the classical universalizing Greek model (the past), while the woman, now in a white dress, scratches and erases the chalkboard, lets her mind wander while sitting on the classroom tables, and walks around, inside and outside, looking at the landscapes. In the meanwhile, we hear a girl singing *Panis Angelicus*,[21] also in a composition made up of fragments and overlays (instead of linear and complete song). Afterwards, the woman's body completely takes over the sequence, looking into the mirror, lying on the bed, walking through Kinshasa's most emblematic monument, the Tour de

l'Échanger, built as a tribute to Patrice Lumumba, looking up, looking outwards—always looking to a possible (future) horizon.

*Beauty* actually seems to encapsulate the whole idea of decolonization as a far-reaching ongoing project, especially if we take into account that "only the beautiful in a strict sense is a specifically aesthetic category that is, in a way, modern" (Carchia and D'Angelo 1999, p. 51) (my translation). Beauty can decolonize the cultural, educational, and representational structures that sustain the lingering effects of colonialism by addressing the main ideas around artistic creation, without recurring to iconoclast or anti-aesthetic visual strategies. Beauty is everywhere and takes on many forms. It is not restricted to wholeness, to linearity, to some sort of idealized perfection. It is a possibility. An envisioned future. In the same way, the project *Panorama* (2017) builds upon the architectural ruins of Portuguese Colonialism, spaces that were once spaces of modernity, of progress, spaces of *whiteness*, spaces of segregation (as this was one of the most sophisticated and elite spaces in Luanda during the colonial occupation), foundations of the construction of the Portuguese colonial project. The project included several series of photographs that depict modernist constructions like Hotel Panorama and Cinema Karl Marx (that was called Cinema Avis before the independence[22]), ruined and abandoned places that are no longer occupied by white bodies. They are now abandoned and taken over by the passage of time, scarred by the many changes and conflicts that occurred since they were built in the 1960s and 1970s (and it is very significant that cinema was one of the many branches of the New State propaganda, idealized by António Ferro[23]). The series *Cinema Karl Marx* (2017) includes a tryptich[24] in which the image of the ruined building is cut into three separate pieces, as if hinting at the continuous splits not just symbolized by this particular construction, but actually inflicted on to (organic or inorganic) bodies (and here I most stress the various semantic dimensions of the word *split*—a separation, a break, a crack, a fissure, but also, in a more metaphorical way, a wound. And these splits are, most of all, political, referring to the ruins of the Colonial Empire, or to the ruination process of the civil war. As de Miranda says: "The cinemas in ruins exist all over Angola because, during the civil war, leisure was not the place. They are, therefore, political places that still tell us lots of stories today" (Macedo et al. 2022, p. 212). In another work of this series, the diptych *Twins*[25] moves to the inside of the cinema, and shows us the ruined auditorium, its chairs empty and broken. Or almost empty: the portrait of two women, twins, dressed in black, sitting with their hands on their lap looking straight forward, not to the screen, but to the camera—to the making of the image and not to the already produced image. "The twins [ . . . ] represent that history of the past and of the present [ . . . ]. [They] refer back to a place of duality and belonging. The fact that they are occupying an empty space with dusty chairs is also an indicator of a reflection on those ruins and of past memories" (Macedo et al. 2022, p. 212).

## 4. On Archival and Material Ruins

*Spell Reel* is part of the project *Luta Ca Caba Inda*, a collaborative art and research project that, although being authored by the Portuguese artist and filmmaker Filipa César, is presented as a collective endeavor, built around the remains of the archive of the National Institute of Cinema and Audiovisual of Guinea-Bissau (INCA). A Creole expression, *Luta Ca Caba Inda* means 'the fight is not over yet', and it is also the title of one of the films rescued from this archive: "We appropriated the title and its spell for a series of public viewings and discursive events used to channel the contents of that fragmented corpus, and we chose to welcome its contradictions", says Filipa César (César et al. 2017, p. 7). So, the project itself included not only the study of the material that composed this archive and its digitalization in Berlin, but also a series of screenings of this same material and several public events that were intended to raise a discussion around the archive, its situation, and many issues arising from it—a process that lasted for about six years. Guinean directors such as Sana Na N'Hada and Flora Gomes participated in this process. They were part of a group that, at the end of the 1960s, was sent by the PAIGC[26] leader Amílcar Cabral to study cinema in Cuba, at the ICAIC (Instituto *Cubano* del Arte e Industria Cinematográficos)—a group

that also included Josefina Lopes Crato and José Bolama Cobumba. Cabral, the leader of the revolution in the territories of Guinea-Bissau and Cape Verde against Portuguese Colonialism who was assassinated in January 1973, just months before the unilateral proclamation of Independence in September of the same year, was well aware that the revolution needed to be recorded, filmed for posterity. The moving image was indeed very important in a social context marked by illiteracy, and, as director Flora Gomes states, Cabral was very much aware of this. As Ros Gray documents, these international movements and relationships of 'African liberationist filmmaking' were not assisted solely by the Soviet Union, but also by Cuba and Yugoslavia. And, as far as Guinean filmmakers' training and Cuba and, after 1972, in Senegal [with Paul Vieyra, from Benin] were concerned, she pointed out that they "signalled an ongoing commitment to technical and pedagogical assistance for the cause of African Liberation from Cuba; the cultural part of a massive military intervention made often in advance of and against the wishes of the Soviet Union" (Gray 2013, p. 55). The film Conakry (Filipa César, Diana McCarthy and Grada Kilomba, Germany, 2012) and the book *Luta ca caba inda: time place matter voice 1967–2017* (ed. Archive Books) also integrate the project.[27]

Launched in, 2017, *Spell Reel* can be said to be a process-film itself, as it is composed of the various dimensions of the project *Luta Ca caba Inda*. In addition to images from the INCA archive and the various films—or rather film fragments found there (due to the poor state of conservation of the films, the deterioration of the material meant that only fragments of the footage shot in the 1970s and 1980s survived)—we can also hear Sana Na N'Hada and Flora Gomes commenting on and, in a certain way, providing a sort of reading or translation of these recorded images. In other moments, there are images of the viewings of the archive footage taken in various regions of Guinea-Bissau and also in Europe, making it a kind of "transnational itinerant cinema" (César et al. 2017, p. 418). We can also see comments by the population to these images or discussions by critics around this material In the same way that *Spell Reel* weaves together past and present by putting together fragments of the cinematic archive, testimony, and uttered voices, so does the book *Luta ca caba inda: time place matter voice 1967–2017*. This book is not just an archive of the project as a whole; it also functions as an ongoing conversation about the ontology of the archive itself through images, texts, documents, references to the material archives, and—this is a very significant aspect—the transcription of sound to the written word, bringing together the affinities that exist between film and book as mediums. As César notes in the conversational prologue between herself, Tobias Hering, and Carolina Rito, "the book introduces a specific dimension to cinema: we can look at still images, at the adhesive tape that joins two film reels, at the pixelated celluloid grains. We can observe other beings sprouting in this misty milieu—the vinegar syndrome and other pathologies" (César et al. 2017, p. 11). The ruin (the material) showing its own process of ruination. As Georges Didi-Huberman observed, "it is extraordinary that men have entrusted so many images, so many affects, so many constructions, such beauty, *to a medium so close, ontologically, to its own ruin*" [cited in Habib 2006, p. 120]. But maybe, just maybe, this is precisely the reason why film archives have been so appealing to artists: because their impending disintegration is what makes them so close to our own mortality as human beings. We are able to engage affectively with these ruined/ruinating bodies because we can see them, and sense them, as our own: *Celluloid grain/Converted into pixels/Pixels converted into fireflies* (*Spell Reel*). One of the objectives of César's project was to digitize the archive (one digitized copy is now in Guinea-Bissau, the other in Berlin, at the Arsenal), but they never intended to restore the footage. The archive was deeply damaged because it was laid out in the streets, abandoned, and literally thrown away because the new government emerging after the civil war in 1998 decided to use the archive's space for other purposes. As Filipa César says, "none of these words—recovering, restoring, preserving—apply to *Luta ca caba inda*, which is about film materials as contemporary agents" (César et al. 2017, p. 165). And that is why the group started calling it a "collective milieu" instead of an archive—the impossibility of the archive once again called for reinvention, it called for a future.

The reinvention of the archive, and of the ruined material, thus means that it needs to be reassembled by imagination and thus brought back to life. Re-imagined. Such is the role of the artist working with the archive. In the book *Devant le Temps. Histoire de l'Art et Anachronisme des Images,* Didi-Huberman argues that art history is an anachronic discipline: "When facing an image", he says, "we are always facing time", and the image always encompasses past and present, memory and future" (Didi-Huberman 2000, p. 21) (my translation). History, therefore, is not exactly "the science of the past", as stated by Marc Bloch, it is an "impure agency", *montage* of time and knowledge, for what the historian does is to "summon and interrogate memory, not exactly «the past»" (Didi-Huberman 2000, p. 39) (my translation). And the *image* is the element that "dismantles history": it takes and breaks something down, disperses, confuses, deconstructs so that something can be built or put together afterward. Therefore, says Huberman, "the historian reassembles the «remains» because they themselves have the dual capacity to dismantle history and to assemble heterogeneous times together. Once with Now, survivals with symptoms, latencies with crises…"" (Didi-Huberman 2000, p. 147). Facing the ruined archive, the artist is confronted with its impossibility (Green 2006), or the impossibility of discursive production from these remains. But it is, nonetheless, put back together through a montage of memories, images, materialities, and affects—an imaginary that converts *celluloid grains into pixels, and pixels into fireflies*. From the decaying body of film, "life" is thus created. But this archive was not only determined by the selection initially made (determining what is rendered visible) and by the ruin of the medium. Another symptom is at play here: its absences and invisibilities. Not being a film about women per se, *Spell Reel* opens with women's stories, with the testimony of a guerrilla woman who speaks about her familiarity with the war arsenal, a testimony that is, in turn, mediated by a man who translates the woman's speech. From an almost deferred voice, this woman becomes a face, a body, but she is not yet a subject. She reappears later on. And the answer to who is this woman emerges in the book *Luta ca caba inda*: her name is Adama Djansane (César et al. 2017, p. 87).

In a commentary to the footage she has seen in Boé in November 2014, Djansane says:

*Let a HK [Heckeler & Hoch] machine gun sound, I recognize it.*

*Let a mortar sound, I recognize it.*

*Let a G3 sound, I recognize it.*

*Let a Mauser sound, I recognize it.*

*Let a canon sound, I recognize it.*

*Let any kind of heavy artillery sound, I recognize them all.*

She was part of the guerrilla.

Josefina Crato is another 'invisibility' that emerges from this work. Not much information is known about her, other than that Crato worked with Sana Na Hada and Flora Gomes in the films *O Regresso de Cabral* and *Anos na Oça Luta* (both made in 1976) and as sound editor, for example in Sarah Maldoror's 1980 documentary *Carnival in Bissau*. Very little is said about her throughout this project, except that she was a part of the group that went to Cuba in 1967—"Josefina Lopes Crato, just to show that one of us was a girl", said Flora Gomes. So, the absence of image of Adama Djansane and the translation of her testimony by a man who gives voice to her is paired with Crato's *lacunae*. But *Spell Reel* recuperates and makes visible one reel of the archive (numbered 019) that contains unedited footage for a film about the situation of women in Guinea-Bissau that was shot between 1977 and 1982. These images were supposed to be a film that was never put together: "unedited footage of women's role in politics, in schools, as workers, as mothers. Masses of women demonstrating in the streets. 1st Congress of Women in Guinea-Bissau and commemoration walk by the UDEMU, Democratic Union of Women of Guinea-Bissau and Cape Verde" (César et al. 2017, p. 259). Also a landmark of this archive is *Un Balcon en Afrique*. A film by Anita Fernandez in collaboration with local filmmakers, it features a white European woman living in a tree house in Bissau and looking down at what was

happening on the ground. "A metaphor for the situation of sympathetic European activists visiting post-independence Africa" (César et al. 2017, p. 281). Anita comments on her plans for the film, and recalls that Sana Na Ha'da and Flora Gomes, collaborating with her on this project, were skeptical about the film, mainly because this was a white protagonist in a high-flying position and because "they felt uncomfortable with the fact that she was a woman" (César et al. 2017, p. 289). Being watched today, this film still raises much discussion. For Grada Kilomba, Fernandez's film brings to the fore "how the emancipation of white feminists took place at the cost of the colonial subject" (César et al. 2017, p. 293). Another lacuna, another ruin, perhaps, not of a colonial past but of a colonial present. One that is still ongoing and operating.

**Funding:** This research received no external funding.

**Data Availability Statement:** Not applicable.

**Conflicts of Interest:** The author declares no conflict of interest.

## Notes

1   A "( . . . ) methodology-pedagogy of conversation", as Walter Mignolo and Catherine E. Walsh refer to it (Mignolo and Walsh 2018, p. 7).

2   Salazar was appointed as Minister of Finance in 1928 and, in 1933, became the President of the Council. As Minister of Finance, he implemented measures of financial rigour that, according to the Minister himself, in his inaugural speech, corresponded to "( . . . ) rigid principles that will guide common work, [and which] show the determined will to regularise for once our financial life and with it the national economic life". The reformulation of the 1911 Constitution took place, years later, in 1933, coming into force on 11 April 1933, consecrating "the Family as a social cell" (Miguel 2007, p. 14). The document stated that the 1930 Colonial Act was also a constitutional matter. Revised in 1935 and 1945, and with four fundamental features: the idea of empire, the greater concentration of powers ( . . . ), the strong demand for national order, and the integration of the colonies and metropolis in the "pluriform unity of the Portuguese Nation" (Rosas et al. 1996a, p. 21).

3   In 2017, the daily newspaper *Público* published a piece titled "Nem o 25 de Abril derrubou o mito do bom colonizador" ("Not even 25 April overturned the myth of the good coloniser"). In the article, journalist Joana Gorjão Henriques writes, "We put racism, slavery and colonialism under the bed. There was no «truth and reconciliation process». The need to decolonise «is not a metaphor». In *Público* 23 September 2017. Available online in https://www.publico.pt/2017/09/23/sociedade/noticia/nem-o-25-de-abril-derrubou-o-mito-do-bom-colonizador-1786395 accessed on 5 December 2022). Researchers Marta Araújo and Silvia Rodríguez Maeso analyze the absence of raciality and political discourse that results in what they term as a "post-colonial-consensus in Portugal" (Araújo and Maeso 2013, p. 145). For more on this issue see also (Araújo and Maeso 2015).

4   As historian Fernando Rosas notes, this was "the longest modern authoritarian experience in Western Europe" (Rosas 1994, p. 10).

5   The figure of the 'father' referred both to the man ruling the house, as well as to the dictator ruling the country. Margarida Calafate Ribeiro analyzed this *trope* in Portuguese society from and after Estado Novo, departing from the works of visual artists Filipa César, Grada Kilomba and Ana Vidigal and the writer Isabela de Figueiredo, author of the novel *Caderno de Memórias Coloniais*, reading these works as "letters to the father/motherland". Maria Luísa Coelho also provided a reading of artist Helena Almeida's work in relation to her own father, Leopoldo de Almeida, a sculptor deeply related to Estado Novo who authored many of the State's commissions of public statutes that had the clear intention of glorifying Portugal's history of 'Descobrimentos'; that is, the glorification of the 'Empire' starting in the 15th and 16th centuries with the maritime expansion to the African and South American continents (Ribeiro 2020).

6   The expression "proudly alone" was used by António de Oliveira Salazar in a speech about the Colonial War made at the inauguration of the Executive Committee of the National Union in 1965. This was also the year of the assassination of General Humberto Delgado, the academic crisis of 1962 (which led to violent repression by the riot police on the students of Coimbra, following the organization of the 1st National Meeting of Students held in the Coimbra Academic Association), and the dissolution of the Portuguese Society of Writers (that year, the society had proposed the Angolan Luandino Vieira as a candidate for the Nobel Prize. Luandino was a member of MPLA, Movimento pela Libertação de Angolana, one of the most important guerrilla organizations operating against the Portuguese army in the Colonial War that took place on Angolan territory).

7   The genesis of these groups goes as far back as 1925, when the Pan-Africanist Congress was held in Lisbon. For more on the Liberation Movements see Rosas et al. (1996b).

8   In Portuguese, "rapidamente e em força". These were the words used by Salazar when addressing the population on national television (RTP) on 13 April 1961. The statement can be seen in the propaganda television show "Angola, decisão de continuar" [Angola, decision to proceed], transmitted on RTP on 27 December 1961. Recording available at https://arquivos.rtp.pt/conteudos/angola-decisao-de-continuar/ (accessed on 5 December 2022).

9  Michelle Salles reflects about the lingering effects of this idea through the work of Yonamine, particularly the installation Tuga Suave [soft Tuga, in a free translation]. This title refers both to the a brand of cigarettes, quite popular in Portugal, Português Suave, and to the deprecating way the Portuguese were referred to by Africans, that became particularly spread during the Independence War because it was commonly used by the guerrilla] that uses popular cutural images (in this case, a pack of cigarettes of the brand Português Suave, quite popular in Portugal) to critically examine the lingering effects of colonialismo, focusing on culture and the language. She also emphasizes that Gilberto Freyre's lusotropicalismo "had been transformed into a fertile field of racist neo-colonial representation in the field of image" (Salles and Lança 2019).

10  News available at https://www.publico.pt/2022/09/02/politica/noticia/primeiroministro-portugues-pede-desculpa-mocamb ique-massacre-wiriamu-2019244 (accessed on 5 December 2022). The Wiriamu Massacre took place in 1972 in Wiriamu village, Mozambique by the hand of the Portuguese colonial military. British journalist Peter Pringle reported the massacre in The Times on 10 July 1973. At least 385 people were killed: it was not "an act of excess of power by some individuals, it was done in obedience to orders from a regime and the Portuguese state. This massacre was planned and executed as planned", historian Mustafah Dadh says (at https://www.publico.pt/2015/11/30/sociedade/noticia/wiriamu-a-vida-antes-e-durante-o-massacre-1715828 accessed on 5 December 2022). For more on the Wiriamu massacre see (Dadah 2016).

11  This book is a part of the ERC-funded project MEMOIRS, coordinated by Margarida Calafate Ribeiro. As we can read on the project's website, "MEMOIRS will offer a radical, alternative and innovative vision of contemporary European history, drawing on the colonial legacies". For more on this project see https://memoirs.ces.uc.pt/?id_lingua=2 (accessed on 6 December 2022).

12  Post-memory, a concept theorized by Marianne Hirsch, refers to the second-generation relationship with traumatic memories, both individual and collective, and its cultural effects (Hirsch 2012).

13  After the Independence, Angola was struck by a civil war between UNITA (of communist allegiance) and FRELIMO (anti-communist) that lasted from 1975 to 2002. This was one of many Cold War battlefields, with both being supported by the USA and the USSR.

14  For example, only quite recently, in 2020, the daughters and sons of undocumented migrants that were living in Portugal had the right to Portuguese citizenship. Many of them never even set foot in their parents' home country, but, despite having been born in Portugal, were not considered by the State as Portuguese. This of course carried many practical problems and limitations to a whole generation—"a generation of Afro-descendants that was born in Portugal but feels like a 'migrant Portuguese'" (Gorjão Henriques 2016)—but it also contributed to a generalized feeling of *unbelonging*. This situation only ended when the 1981 Law of Nationality (Law n° 37/81, 3 October) was changed in 2020 (Organic Law n° 2/2020, 10 November) (by which someone born in Portugal could obtain Portuguese citizenship if one of their parents lived in Portugal legally, or in an illegal situation for more than one year. For more on the issues of identity and post-coloniality in the Portuguese context see Ribeiro Sanches (2006)).

15  Writer Djaimilia Pereira de Almeida, born in Luanda and living in Portugal since a very young age, portrayed the many entanglements of African migrants in Portugal (living mainly in Lisbon) in brilliant narratives that weave political, geographical, and identity issues around African characters that have one essential characteristic—mobility. Esse Cabelo (de Almeida 2015) and Luanda, Lisboa, Paraíso (de Almeida 2018) are particularly compelling narratives. About the latter, Margarida Calafate Ribeiro writes: "What is at stake in the book by Djaimilia Pereira de Almeida are the living and human ruins of the empire, no longer from the figure of the ex-combatant, nor of the returnee, but of those who were on the other side of the line that colonialism drew: the blacks and, in this case, the most complex figure that colonialism generated, the assimilated that for the first time in Portuguese literature is at the centre of the narrative" (my translation) (Ribeiro 2019).

16  HD video, color, sound, 22´ triptych screen; Director and Editing: Monica de Miranda; Camera: Tiago Mata Angelino; Performance: Andre Cunha, Bruno Giordani, João Silva, Gil Rodrigues, Mónica de Miranda. Available at https://monicademiranda.or g/videos/once-upon-a-time-video/ (accessed on 25 November 2022).

17  These ruined ships are the perfect visual metaphor for the intricate dependency of colonialism and capitalism, central also in the slave trade between Africa, Europe and South-America.

18  Mónica de Miranda cited in Jayawardane (2018).

19  Considering its beginning in the Renaissance period.

20  HD vídeo, sound 6´´ with sound design by Soundslikenuno (Chullage). Available at https://monicademiranda.org/videos/bea uty/ (accessed on 25 November 2022).

21  "The Bread of Angels" was written by Saint Thomas Aquinas for Corpus Christi in the 13th Century with the lyrics: Panis angelicus/Fit panis hominum/Dat panis coelicus/Figuris terminum/O res mirabilis!/Manducat Dominum/Pauper, pauper/Servus et humilis.

22  Inaugurated in 1961, the project was conceived by two Portuguese architects, Luís Gracia de Castilho and João Garcia de Castilho.

23  Director of SPN—Secretariado de Propaganda Nacional (Secretariat of National Propaganda, created by the regime in 1933). A modernist, Ferro's dance group *Verde Gaio* (inspired by Sergei Diaghilev's *Ballets Russes*) promoted the production of national cinema based on comedies and organized the Exposição do Mundo Português (Exhibition of the Portuguese World 1940). According to Artur Portela, Salazar's SPN was different from that of Ferro: while he first saw it as a "collaboration of the greatest Portuguese values" (Portela 1987, p. 23), for Ferro, in his own words, the SPN's function was to promote Salazar and the Estado Novo, "the new, the most advanced impulse, the avant-garde" (cited in Portela 1987, p. 25).

24  *Cinema Karl Marx*, 2017. Inkjet print on fine art paper. 100 × 249 cm. Available at https://monicademiranda.org/photography/panoram-2/ (accessed on 25 November 2022).

25  Inkjet print in fine art paper. 60 × 90 cm. Available at https://monicademiranda.org/photography/panoram-2/ (accessed on 25 November 2022).

26  Partido Africano para a Independência da Guiné-Bissau e Cabo Verde (African party for the Independence of Guinea and Cape Verde).

27  Carried out in partnership with Arsenal—Institute of Cinema and Video Art, Berlin, Jeu de Paume, Paris, The Showroom, London and ZDB, Lisbon.

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
