# Peer review of "Around Ruins: Some Notes on Feminist and Decolonial Conversations in Aesthetics"

_arts, 2023_

Round 1

Reviewer 1 Report

In general the text is well researched, particularly in regards to theory and the methodologies of theory. My main concern though is that there is more theory than actual art discussion.  Although I recognize the artists have a theoretical interest, or theory is being used to understand their practices, there is a more developed discussion on theory than on the artists themselves. The art that is mentioned at the beginning is not as fleshed out as the theoretical stand points.  

My suggestion, and others my have offered more productive avenues here, would be consider to synthesize a bit more on the historical and political context (I find it quite compelling, but is it all necessary for a text of this length?); and a bit more condensing of the theory; a challenge I know.  This would open a bit more room for the discussion of the art and the artists themselves. I think the essay would benefit as well with a better conclusion/discussion section.  

I will say this topic itself has real merit and brings interesting questions to the discourse. 

Reviewer 2 Report

Well written and good research. 

Reviewer 3 Report

I found this very useful indeed. The article only claims to be "notes", but it is thorough in its referencing and there is clear movement towards elaborating the notion of "ruin."  I found much to think about in relation to temporality and the archive, and to the positioning of sympathetic white feminists in the elaboration of a genuinely interracial analysis.

I hope the author might like to use this piece as the basis for a book, especially if it were to expand its reference to C20 feminist aesthetics as well as the more recent. There is much to explore here: eg in C20, Elizabeth Grosz, later work of Judith Butler; and C21 moves towards scientific disciplines.

Author Response

Thanks for your reviewing. See the attachment
